# Predicting lymphatic filariasis elimination in data-limited settings: A reconstructive computational framework for combining data generation and model discovery

**Morgan E. Smith**[1], **Emily Griswold**[2], **Brajendra K. Singh**[1], **Emmanuel Miri**[3], **Abel Eigege**[3], **Solomon Adelamo**[3], **John Umaru**[3], **Kenrick Nwodu**[3], **Yohanna Sambo**[3], **Jonathan Kadimbo**[4], **Jacob Danyobi**[5], **Frank O. Richards**[2], **Edwin Michael**[1] *

**1** Department of Biological Sciences, University of Notre Dame, Notre Dame, Indiana, United States of America, **2** The Carter Center, One Copenhill, Atlanta, Georgia, United States of America, **3** The Carter Center Nigeria, Jos, Nigeria, **4** Plateau State Ministry of Health, Jos, Plateau, Nigeria, **5** Nasarawa State Ministry of Health, Lafia, Nasarawa, Nigeria

\* emichael@nd.edu

**Data Availability Statement:** All relevant data are within the manuscript and its Supporting

## Abstract

Although there is increasing importance placed on the use of mathematical models for the effective design and management of long-term parasite elimination, it is becoming clear that transmission models are most useful when they reflect the processes pertaining to local infection dynamics as opposed to generalized dynamics. Such localized models must also be developed even when the data required for characterizing local transmission processes are limited or incomplete, as is often the case for neglected tropical diseases, including the disease system studied in this work, viz. lymphatic filariasis (LF). Here, we draw on progress made in the field of computational knowledge discovery to present a reconstructive simulation framework that addresses these challenges by facilitating the discovery of both data and models concurrently in areas where we have insufficient observational data. Using available data from eight sites from Nigeria and elsewhere, we demonstrate that our data-model discovery system is able to estimate local transmission models and missing pre-control infection information using generalized knowledge of filarial transmission dynamics, monitoring survey data, and details of historical interventions. Forecasts of the impacts of interventions carried out in each site made by the models estimated using the reconstructed baseline data matched temporal infection observations and provided useful information regarding when transmission interruption is likely to have occurred. Assessments of elimination and resurgence probabilities based on the models also suggest a protective effect of vector control against the reemergence of LF transmission after stopping drug treatments. The reconstructive computational framework for model and data discovery developed here highlights how coupling models with available data can generate new knowledge about complex, data-limited systems, and support the effective management of disease programs in the face of critical data gaps.

Information files. Code is available at https://github.
com/EdwinMichaelLab/EPIFIL_hindcast.

**Funding:** The author(s) received no specific
funding for this work.

**Competing interests:** The authors have declared
that no competing interests exist.

## Author summary

As modelling becomes commonly used in the design and evaluation of parasite elimination programs, the need for well-defined models and datasets describing the nature of transmission processes in local settings is becoming pronounced. For many neglected tropical diseases, however, data for site-specific model identification are typically sparse or incomplete. In this study, we present a new data-model computational discovery system that couples data-assimilation methods based on existing monitoring survey data with model-generated data about baseline conditions to discover the local transmission models required for simulating the impacts of interventions in typical endemic locations for the macroparasitic disease, lymphatic filariasis (LF). Using data from eight study sites in Nigeria and elsewhere, we show that our reconstructive computational framework is able to combine information contained within partially-available site-specific monitoring data with knowledge of parasite transmission dynamics embedded in process-based models to generate the missing data required for inducing reliable locally applicable LF models. We also show that the models so discovered are able to generate the intervention forecasts required for supporting management-relevant decisions in parasite elimination.

## Introduction

The knowledge discovery, prediction, and forecasting capabilities offered by mathematical models are making these tools central to the effective design and management of long-term parasite control and elimination programs [1–6]. Growing collaborative efforts between modelers and policy makers through channels such as the Neglected Tropical Disease (NTD) Modeling Consortium highlight this critical role of parasite transmission models for supporting global health policy development and motivate their continued refinement [2]. This work has shown how these models are useful for their ability to synthesize diverse sets of information, to evaluate the likely outcomes of what-if scenarios that outline management options, and to discover new knowledge about transmission and elimination dynamics [6–8].

However, there is also increasing recognition that parasite transmission models are useful for management only if they can represent locality-specific infection dynamics adequately and are able to generate reliable predictions of applied interventions in a particular setting [9–12]. This requirement implies that these models should accurately reflect the key biological and ecological processes underpinning local parasite transmission through the reliable specification of initial conditions, parameterizations, and the functional forms describing specific transmission processes whose uncertainties are well-defined [13–15]. A growing theme is also how best to discover or infer these input and latent variables from external observations so that information contained within data can be reliably captured for mathematically reconstructing local transmission dynamics [13,16–18]. This requirement suggests that improving model predictive performance will ultimately depend on the development and application of computational discovery systems that can construct suitable models from process information contained in empirical data [19].

Data assimilation methods aim to improve model specification and performance by explicitly combining them with data such that models are suitably learned from empirical observations [13,15,17]. Such statistical techniques are widely used in ecological modeling to constrain parameters and state variables to match reality as closely as possible before simulating out-of-sample conditions [13,17]. Indeed, in our previous work we have demonstrated the value of

data assimilation for improving knowledge of and for predicting the transmission and elimination dynamics of the vector-borne macroparasitic disease under study here, viz. lymphatic filariasis (LF), a highly debilitating disease which is targeted for global elimination by the World Health Organization (WHO) [9,10,20,21]. This work has demonstrated how, in addition to improving model accuracy, data assimilation can also significantly decrease the uncertainty associated with model predictions, highlighting a clear benefit of using these approaches for policy applications that require both accuracy and precision [13,16,18,22].

A significant challenge in the ability to generate locally relevant predictions for LF (and indeed all NTDs), however, is that the data available for the identification of location-specific models is often sparse, incomplete, or missing. A major concern is uncertainty pertaining to initial conditions because model predictability, even in the case of deterministic systems, are highly sensitive to the values of the inputs describing the endemic disease state [14,23]. Previous LF modeling studies, for example, suggest that transmission thresholds, timelines to achieve elimination, and the likelihood of infection reemergence all vary significantly according to the undisturbed pre-control prevalence and mosquito biting rate [9–11,20,24,25]. Knowledge of baseline conditions is also essential for predicting trajectories of infection prevalence under pressure from interventions for assessing whether programs are progressing as expected [4,5]. These considerations imply that modeling exercises that do not adequately account for observed baseline conditions could have very limited forecast horizons and risk making misguided predictions that undermine their utility for policy.

An example of a program facing critical data gaps is the LF elimination program in Nigeria. The Carter Center (TCC), in partnership with the Nigerian Ministry of Health and international donors, began administering their LF control (and eventually elimination) program in 2000 focusing on mass drug administration (MDA) using ivermectin (IVM) and albendazole (ALB). Surveys were conducted to monitor and evaluate progress of the program at the start of the LF program (1999–2000) and generally every year thereafter in representative sentinel villages (SVs) (a key geographical population entity used for program monitoring [26]). However, some Nigerian SVs are co-endemic for onchocerciasis and thus have unique prior histories with MDA. TCC began preventative IVM+ALB treatments for LF after many years of IVM monotherapy had already taken place for onchocerciasis. Because IVM also works against *Wuchereria bancrofti* (the parasite responsible for LF in Nigeria), the "baseline" LF surveys do not reflect the true pre-control prevalence. For areas such as these SVs, where there is no historical data to guide the setting of initial boundary conditions, new computational approaches that can mathematically reconstruct the prevailing transmission dynamics in a setting by generating both models and missing observations will be required to make useful predictions.

Here, our aim is to develop and investigate the utility of a novel reconstructive computational system to facilitate both the discovery of data and models in order to generate predictions pertaining to LF elimination in areas where we do not have sufficient observational data for model calibration. Our approach to overcoming the observed data limitations is essentially to incorporate domain-specific knowledge of the system dynamics, so-called "common sense" constraints, into the computational framework [8,27], and use the constrained models that are identified using existing data for generating the key data that were previously unknown. The consequent information about the local transmission system discovered by assimilating process information contained in the generated and observed data into the mathematical model is then employed for simulating the impacts of interventions [19].

Our discovery system has the following steps. First, we modified our previously developed Bayesian Melding (BM) framework to estimate local LF models for eight study sites in order to hindcast the unknown pre-control LF infection prevalence in each site. The modified calibration approach relied on the mechanistic understanding of parasite transmission dynamics

represented in our general LF simulation model, information about historical treatments, and longitudinal monitoring data collected during control programs [21,28]. The datasets employed for carrying out this exercise represent five TCC-supported SVs in Nigeria and an additional three SVs from other global endemic regions. We then use the estimated site-specific models along with the hindcasted baseline information to 1) calculate location-specific transmission thresholds and 2) to simulate the impact of MDA and vector control (VC) interventions in each setting based on observed or estimated population coverages. Five of the eight study sites had pre-control data available and thus served as validation cases from which we determined the relative information loss associated with calibrating the model without baseline data. The results indicate that there is a trade-off between predictive performance and uncertainty in initial conditions, but, importantly, that the predictions were still useful for program management purposes. Overall, the reconstructive computational discovery system developed in this study points to how a modelling platform that allows the complementary identification of mathematical models and data can serve as a vital tool for addressing knowledge gaps and making policy relevant predictions even in the face of significant uncertainties and missing data.

## Methods

### Study site descriptions

The application of the computational framework developed in this study is dependent on the characteristics of the observational data available for a given study site. For demonstrative purposes, longitudinal epidemiological data collected by TCC from five SVs in Plateau and Nasarawa states in Nigeria were selected for this modelling study based on data availability and overall representativeness of endemicity and control history. Three additional sites from Tanzania, Egypt, and Papua New Guinea (PNG) were also included to introduce diversity in the transmission settings [29–33]. Details regarding survey methodologies and data aggregation are given in S1 File.

In central Nigeria, LF is caused by the parasite *Wuchereria bancrofti*, carried by *Anopheles gambiae* and *An. funestus* mosquitoes [34]. Surveys were conducted to monitor and evaluate progress of the program at baseline (1999–2000) and generally every year thereafter in the SVs. SVs in Nasarawa and Plateau states were selected based on their representativeness in terms of baseline prevalence, onchocerciasis co-endemicity (or non-endemicity), and prior experience with MDA. Village residents and leadership also had to be willing to participate in annual nocturnal blood surveys for an indefinite period of time. Both serologic (circulating filarial antigen (CFA) as determined by immunochromatographic test (ICT)) and parasitologic (nocturnal blood slides stained and read for microfilaremia (mf)) indicators were monitored annually.

Three of the Nigerian SVs considered in this study (Seri, Gbuwhen, and Maiganga) are co-endemic for onchocerciasis and thus have unique histories with MDA. TCC began preventative IVM+ALB treatments for LF in Gbuwhen in 2000 and in Seri and Maiganga in 2001 following 6–9 years of IVM monotherapy for onchocerciasis (Table 1). Because IVM also works against *W. bancrofti*, the baseline LF surveys do not reflect the pre-control prevalence. To model transmission in these SVs, it was therefore necessary to hindcast the probable pre-control conditions using our reconstructive computational approach based on model calibration using only mid-MDA data. An outline of the overlapping onchocerciasis and LF activities are given in Table A in S1 File for Seri as an example of this situation.

We used the other two Nigerian SVs (Dokan Tofa and Piapung) plus the three global sites (Kirare, Tanzania, Giza, Egypt, and Usino Bundi, PNG) to evaluate the capability of our methodology to hindcast endemic conditions. Unlike the previously described settings, each of

**Table 1. Monitoring epidemiological survey data and intervention details for Seri, Gbuwhen, and Maiganga, Nigeria.**

| Village | Seri | | | Gbuwhen | | | Maiganga | | |
|---|---|---|---|---|---|---|---|---|---|
| Year | % mf prevalence (no. sampled) | % CFA prevalence (no. sampled) | MDA Coverage (% population) (regimen)[1] | % mf prevalence (no. sampled) | % CFA prevalence (no. sampled) | MDA Coverage (% population) (regimen)[1] | % mf prevalence (no. sampled) | % CFA prevalence (no. sampled) | MDA Coverage (% population) (regimen)[1] |
| 1992 | - | - | N/A | - | - | 52.4 (I) | - | - | 70.3 (I) |
| 1993 | - | - | N/A | - | - | 47.8 (I) | - | - | 85.8 (I) |
| 1994 | - | - | N/A | - | - | 41.8 (I) | - | - | 85.6 (I) |
| 1995 | - | - | 77.1 (I) | - | - | 67.5 (I) | - | - | 89.5 (I) |
| 1996 | - | - | 77.7 (I) | - | - | 89.1 (I) | - | - | 94.6 (I) |
| 1997 | - | - | 79.3 (I) | - | - | 85.8 (I) | - | - | 77.4 (I) |
| 1998 | - | - | 79.1 (I) | - | - | 51.9 (I) | - | - | 84.6 (I) |
| 1999 | - | - | 68.4 (I) | - | - | - (I) | - | - | 73.6 (I) |
| 2000[2] | - | - | 75.2 (I) | - | - | 68.4 (I+A) | - | - | 74.1 (I) |
| 2001 | - | - | 74.1 (I+A) | - | - | 62.7 (I+A) | - | - | 85.3 (I+A) |
| 2002 | - | - | 76.7 (I+A) | - | - | 49.6 (I+A) | - | - | 81.6 (I+A) |
| 2003[3] | 10.6 (527) | 33.7 (528) | 78.2 (I+A) | 3.7 (508) | 13.5 (510) | 98.7 (I+A) | 4.8 (478) | 15.2 (587) | 79.8 (I+A) |
| 2004 | 11.3 (97) | 22.4 (474) | 79.3 (I+A) | 1.8 (446) | 5.2 (446) | 93.9 (I+A) | 5.3 (286) | 15.3 (-) | 82.8 (I+A) |
| 2005 | 1.6 (321) | 13.0 (150) | 78.1 (I+A) | 0.3 (286) | 4.5 (178) | 96.6 (I+A) | 3.0 (169) | 20.6 (165) | 90.5 (I+A) |
| 2006 | 1.3 (157) | - | 75.9 (I+A) | 0.5 (183) | - | 92.5 (I+A) | 5.6 (126) | - | 91.5 (I+A) |
| 2007 | 0.8 (133) | 22.6 (385) | 87.4 (I+A) | 0.0 (197) | 6.4 (127) | 91.8 (I+A) | 0.6 (158) | 19.2 (109) | 92.8 (I+A) |
| 2008 | 2.7 (110) | 8.2 (110) | 75.0 (I+A) | 0.0 (127) | 0.0 (175) | 93.9 (I+A) | 1.8 (109) | 1.8 (152) | 72.9 (I+A) |
| 2009 | 0.0 (258) | 10.5 (258) | 85.1 (I+A) | 0.0 (175) | 0.6 (-) | 79.3 (I+A) | 0.6 (152) | 7.9 (-) | 89.7 (I+A) |
| 2010 | 1.1 (268) | 12.7 (268) | - (I+A) | 0.0 (-) | 1.0 (-) | - (I+A) | 0.0 (-) | 11.7 (-) | - (I+A) |
| 2011 | 0.0 (211) | 3.3 (211) | - (I+A) | 0.0 (-) | 0.8 (-) | - (I+A) | 0.0 (-) | 3.0 (-) | - (I+A) |
| 2012 | - | - | 78.7 (I+A) | - | - | - (I+A) | - | - | - (I+A) |
| 2013 | - | - | 79.7 (I) | - | - | - (I) | - | - | - (I) |
| 2014 | - | - | 79.7 (I) | - | - | - (I) | - | - | - (I) |
| 2015 | - | - | 79.8 (I) | - | - | - (I) | - | - | - (I) |
| 2016 | - | 0.0 (74) | - (I) | - | - | - (I) | - | - | - (I) |

N/A not applicable;—not available

[1] I: ivermectin, A: albendazole

[2] Baseline mapping surveys for LF were conducted in 2000

[3] First instance of age-stratified monitoring data, see Table 6 in S1 File

these sites are endemic for LF only and have baseline data that represent the true pre-control prevalence suitable for facilitating validations of the model-estimated baseline infection data. Both mf and CFA were monitored in each site throughout the MDA programs (Tables B and C in S1 File).

Insecticide-treated nets (ITNs) and long-lasting insecticide treated nets (LLINs) were distributed periodically in central Nigeria throughout the 2000s by the local government and international bodies like TCC. ITNs were introduced in Seri in 2004 and LLINs were distributed in all five Nigerian SVs in 2010. Bednet coverage was assumed to be 80.4% for Seri and 50% for the other four Nigerian SVs based on program estimates. Early evidence suggests that these interventions are indeed synergistic in central Nigeria [35–37]. In PNG, the vector responsible for transmission is *An. Punctulatus*, and we assumed LLIN coverage of 40% in Usino Bundi based on regional bednet usage statistics [38]. ITN usage in Kirare, Tanzania was estimated to be much lower, around 25%, as large scale distribution did not occur until later in the control progam in this region [30]. In Kirare, transmission was predominantly mediated

by *An*. g*ambiae* and *An. funestus* during pre-MDA surveys [30]. We therefore assumed *Anopheles* transmission for modeling purposes, but note that *Culex quinquefasciatus* was also present and that species composition studies indicated that the *Culex* mosquitoes were dominant by the end of the MDA program [30]. *Culex pipiens* mosquitoes are responsible for transmission in Giza, Egypt and bednets are not prevalent [32].

## Computational system for data and model discovery

Here, we outline the workflow underpinning our novel reconstructive computational discovery framework for generating local infection data and models simultaneously (Fig 1). We begin by noting that predictions arising from dynamical models, even in the case of deterministic models, are invariably highly dependent on initial conditions [14,23], and thus, in the absence of empirical data describing baseline infection prevalence, it is crucial that we first accurately estimate this key information if we are to simulate control and elimination dependably. A computational discovery system must therefore be capable of facilitating the generation of such missing data by taking advantage of the knowledge contained within process models fitted to training datasets. Here, we accomplished this step by adapting our previously developed Bayesian Melding (BM) data assimilation methodology that originally was used to identify an ensemble of locally acceptable models by calibration of model parameters (Table D in S1 File) to baseline site-specific infection data collected by control programs [9,20,21,39–41]. In the absence of such baseline data, we modified the approach to first calibrate the model parameters using post-intervention mf and CFA monitoring data collected during MDA programs (see calibration steps 1–3 in Fig 1). Then, using the fitted models, we reconstructed new information on the pre-control prevalence before proceeding to predict the relevant quantities

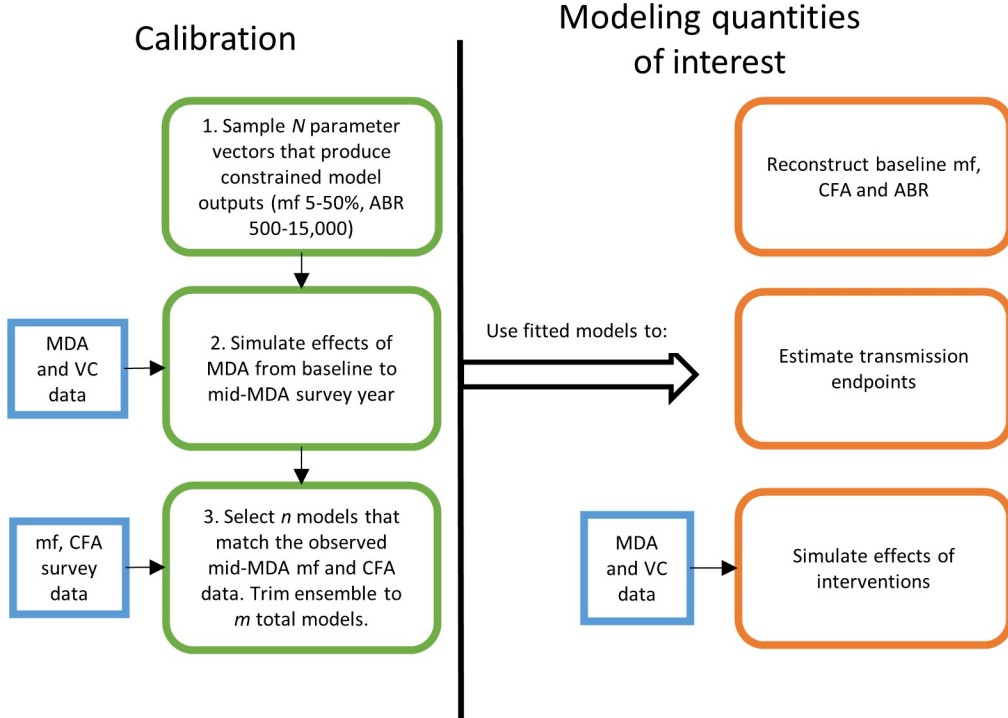

**Fig 1. Computational system for discovering local infection data and transmission models.** Green boxes represent model calibration steps, orange boxes represent the quantities of interest calculated by the fitted model, and blue boxes indicate the site-specific data needed for the indicated step.

of interest, including estimating transmission endpoints and performing simulations of the expected impacts of interventions programs (see quantities of interest in Fig 1). Fig 1 summarizes the calibration (or model discovery), data generation, and modelling steps involved in applying the computation system, while also highlighting the site-specific training and intervention data needed to carry out each of these steps. Full details of the mathematical model, calibration procedures, and methods for calculating these variables are as given below.

**Mathematical model of LF transmission.**   The mathematical model used in this study describes the population dynamics of LF infection through a series of coupled differential equations representing the immigration-death processes of parasite life stages in the human and vector hosts. The technical details of the model and its implementation have been previously explained in detail elsewhere [9]. The state variables representing parasitic infection in the human host are given as coupled partial differential equations representing pre-patent worm burden ($P$), patent worm burden ($W$), microfilariae density in 1 mL blood ($M$), circulating filarial antigen intensity ($A$), and a measure of immunity ($I$), each of which are integrated over age of the human host ($a$) and time ($t$). A single state variable describing the L3 larvae density in the mosquito host ($L$) is formulated as an ordinary differential equation integrated over time ($t$), and this stage is assumed to reach equilibrium quickly ($L^*$) owing to the faster time scale of infection dynamics in the vector compared to the human. The state equations are given below and all model parameters and full functional forms are given in Tables D and E in S1 File.

$$\frac{\partial P}{\partial t} + \frac{\partial P}{\partial a} = \lambda \frac{V}{H} h(a)\Omega(a,t) - \mu P(a,t) - \lambda \frac{V}{H} h(a)\Omega(a,t-\tau)\zeta$$

$$\frac{\partial W}{\partial t} + \frac{\partial W}{\partial a} = \lambda \frac{V}{H} h(a)\Omega(a,t-\tau)\zeta - \mu W(a,t)$$

$$\frac{\partial M}{\partial t} + \frac{\partial M}{\partial a} = \alpha s\phi[W(a,t),k]W(a,t) - \gamma M(a,t)$$

$$\frac{\partial I}{\partial t} + \frac{\partial I}{\partial a} = W_T(a,t) - \delta I(a,t)$$

$$\frac{\partial A}{\partial t} + \frac{\partial A}{\partial a} = \alpha_2 W(a,t) - \gamma_2 A(a,t)$$

$$\frac{dL}{dt} = \lambda\kappa g \int \pi(a)(1 - f[M(a,t)])da - (\sigma + \lambda\psi_1)L$$

$$L^* = \frac{\lambda\kappa g \int \pi(a)(1 - f[M(a,t)])da}{\sigma + \lambda\psi_1}$$

**Initializing the model in the absence of baseline data.**   First, uniform prior distributions were defined for each of the model parameters (Table D in S1 File). Reasonable boundaries on the baseline variables to be hindcasted were also pre-determined. We considered 5–50% mf prevalence and 500–15,000 bites per person per year to be plausible values in this regard [34,42–46]. The model was then initialized by repeatedly sampling parameter vectors (each consisting of a random draw of each of the model's parameters) until a set of $N = 200,000$ parameter vectors that met the following criteria was sampled: 1) the distribution of

community mf prevalence values predicted by the parameter vectors was uniform, and 2) the output mf prevalence values ranged from 5–50%.

**Pass/fail model selection based on mid-MDA data.** Then, using the model parameterized with each of the $N$ parameter vectors, we simulated the observed site-specific interventions (see intervention modelling details below), resulting in $N$ trajectories of mf and CFA prevalence over time and age. The $N$ Monte Carlo simulations were resampled to select only those models that were considered behavioral as determined by a pre-defined binary (pass/fail) criteria of fit described below [41]. This pass/fail resampling approach was previously described for our model in Smith et al. (2017) for a single model output (mf prevalence) [40]. Here, we apply the same criterion of fit jointly to both mf and CFA indicators as follows. The $N$ model-generated mf and CFA prevalence trajectories were judged against the observed follow-up prevalence data from a single mid-MDA survey. Parameter vectors were retained only if the corresponding age-stratified mf and CFA prevalence for the given survey year fell within the 95% binomial confidence intervals of the age-stratified data for more than $j$-2 data points (where $j$ is the number of age groups for which there is prevalence data). This step produced a reduced set of $n$ parameter vectors. Formal goodness of fit of discovered models for data was evaluated by calculating the Monte Carlo $p$-value as described in [21].

The mid-MDA survey year used for model resampling varied by site according to the characteristics of each unique dataset, and age-stratified data was prioritized when choosing a survey. For Seri, Gbuwhen, and Maiganga, the prevalence data from 2003 was used for model selection because age-stratified data was available for both indicators (Table F in S1 File). No age-stratified data from a mid-MDA survey was available in the other sites, so an age profile was derived from the overall prevalence data and national age demographic information as described previously [40]. In these sites, the mid-MDA survey used for resampling was selected based on the following criteria: at least two rounds of MDA must have had already occurred, data was required for both mf and CFA indicators, the prevalence values must have been greater than 2%, and there needed to remain at least two later surveys in the dataset that could be used for model validation. For Dokan Tofa and Piapung, the first survey year that met this criteria data was used (2007 and 2005, respectively). For Kirare, Giza, and Usino Bundi, the surveys from years four, three, and two were selected, respectively.

Once a subset of $n$ models had been selected, we further trimmed the ensemble of models to improve computational efficiency and predictive performance [47]. We grouped the $n$ selected models according to hindcasted baseline mf and CFA prevalence into bins at 5% intervals (5–10%, 10–15%, etc.) and calculated the relative contribution of each bin to the overall ensemble (i.e. the proportion of models falling in a given interval). If any bin contributed less than 10% of the total number of models to the ensemble, the models falling in that bin were discarded as the least likely to represent the baseline conditions. This final set of $m$ site-specific models was used to hindcast the pre-control baseline conditions and then calculate desired model outputs such as transmission breakpoints and timelines to achieve elimination.

**Calculating transmission breakpoints.** A previously developed numerical stability analysis for calculating site-specific transmission breakpoints was conducted for each of the $m$ posterior parameter vectors, producing a distribution of mf breakpoints and threshold biting rates (TBRs) for each site [9,20]. The thresholds are dependent on the mosquito biting rate which can range from the prevailing ABR that has given rise to endemic conditions to zero (theoretically) in the presence of vector control (VC). Because most of the study sites used VC, we calculated the breakpoints at both the ABR and the model-estimated TBR. The target endpoint used to calculate elimination timelines represents a 95% probability of elimination and was derived from the empirical inverse cumulative density function describing the site-specific mf breakpoint distribution.

**Modelling MDA and vector control interventions.** The model formulation for simulating MDA and bednets has been previously described in full and is presented in the S1 File [9]. MDA with IVM and IVM+ALB was modeled as having an instantaneous killing effect on adult worms and microfilariae ($\omega$ and $\varepsilon$, respectively) as well as sterilizing adult worms for a period of months (*p*). Because there lacks strong data for $\omega$, this parameter was sampled from a uniform prior along with the other model parameters (Table D in S1 File). VC by ITNs and LLINs deters mosquitoes from entering the home, prevents them from feeding on humans, and kills them [48,49], the effects of which were modeled by dynamically reducing the biting rate according to the ITN and LLIN coverage and efficacy.

We modeled the observed MDA and VC interventions in each site and compared the predictions to the observed survey data. A survey of MDA coverage in Nigeria suggested that the recorded coverage could be higher than actually achieved [34]. Therefore, the coverage modeled for each round of MDA was sampled from a uniform distribution representing 75–100% of the reported population coverage. Where data was not available, we assumed a value for MDA coverage equal to the average of the population coverage achieved in the previous rounds.

**Calculating elimination and resurgence probabilities.** In addition to the observed interventions, we also modeled hypothetical scenarios where MDA interventions were stopped at either the WHO-recommended 1% mf threshold or at the model-estimated mf breakpoint. The observed MDA coverage was applied for each round. In cases where additional rounds of MDA beyond those that were actually observed were needed to reach the given thresholds, annual MDA was continued at a coverage of 65% (the minimum coverage required by the WHO [26]). After stopping MDA interventions, we continued the simulations with VC only for an additional 10 years to estimate the prevalence during and after the requisite 5-year surveillance period. We then calculated elimination and resurgence probabilities in each site. Elimination was considered to have been achieved by those model trajectories that predicted mf prevalence would continue to decline in the years following the end of surveillance (indicated mathematically by a negative slope between five years and 10 years after stopping MDA). Conversely, resurgence was considered to be expected by those model trajectories that predicted mf prevalence would increase in the years following surveillance (indicated by a positive slope).

## Model performance

For sites where baseline infection data was available (Dokan Tofa, Piapung, Kirare, Giza, and Usino Bundi), the model was also calibrated directly to site-specific baseline infection data using our Bayesian Melding data assimilation framework [9]. Results from these models served as a comparison to the hindcasted models in order to evaluate the performance of the latter models. The model calibration steps are notably simpler in this case. First, $N = 200,000$ parameter vectors were randomly drawn from their uniform priors and used to estimate baseline age stratified mf and CFA prevalence. Then, these model outputs were judged against the observed baseline mf and CFA data using the binary criteria of fit as described above. This ensemble of models was then used to simulate the impact of interventions and produce estimates of interest such as transmission endpoints and timelines to achieve elimination for a given site.

The predictions from the hindcasted models were compared to the predictions from the models fitted directly to baseline data to quantify the trade-offs associated with calibrating the models without baseline data. The relative information lost as a result of missing baseline data was defined as $I = H_b - H_h$ where $H$ measures the entropy, or uncertainty, associated with the baseline mf prevalence estimates, $H_b$ denotes predictions from the model fitted directly to

baseline data, and $H_h$ signifies predictions from the hindcasted model fitted to mid-MDA data. The Shannon information index was used to measure entropy, $H$, as follows: $H = -\sum_{i=1}^{m}$ $p(x_i)log_2p(x_i)$, where $p(x_i)$ is the discrete probability density function of the baseline mf prevalence predicted by the given model, and is technically estimated from a histogram of the respective model predictions for $k$ bins (of equal width in the range between the minimum and maximum values of the PDFs) [22,50]. A smaller value of $H$ indicates less uncertainty.

## Results

Using the computational framework described above, we aimed to learn new knowledge about local LF transmission systems through the combination of data and model discovery. To overcome the lack of baseline data, we developed a new approach that provides flexibility for both hindcasting and forecasting by applying domain knowledge and common sense constraints where key data was missing. We evaluated the framework's ability to fill in data gaps, specifically hindcasting the unknown baseline conditions, and predict the impacts of interventions. We further used the fitted models for estimating variables pertinent to elimination programs such as transmission breakpoints and elimination and resurgence probabilities.

### Validating the computational framework for hindcasting baseline conditions

To validate the new approach for calibrating the model without baseline data, we applied the Bayesian Melding framework described above [9] to the five sites that have pre-control data available for comparison (Dokan Tofa, Piapung, Kirare, Giza, and Usino Bundi). Note that we modeled these sites assuming the baseline data was not available, i.e. the model was not given this information. The model fits to the training mid-MDA mf and CFA survey data are given in Fig A in S1 File and the corresponding hindcasts of the likely baseline conditions are shown as age-prevalence curves in Fig 2. Note that since age-stratified infection profiles were derived from the overall community prevalences estimated in these sites, some uncertainty is introduced into these hindcasts. However, comparison of the model hindcasts against the respective observed baseline infection data show that the generated hindcasts capture the observed data in each site well (overall Monte Carlo $p$-values >0.9, indicating no significance difference with the data). Other than CFA in Usino Bundi, the ensemble of model hindcasts overlaps with all of the observed data points and their 95% confidence intervals.

We also compared the predictions coming from the models fit using the new computational approach in the absence of baseline to those coming from models fit directly to baseline data. This was done using the data from the five sites that had pre-control infection data available. Entropy calculations indicate that information is lost by fitting the model without baseline data (Table 2). However, the model fits to baseline data (Fig B in S1 File) are visually similar to the hindcasted baseline estimates (Fig 2). The breakpoint distributions are also similar between the two fitting procedures (Fig C in S1 File) and Kruskal-Wallis tests indicate that there is not always a statistically significant difference in breakpoint distributions (Table G in S1 File). Similarly, the median predicted timelines to reach 1% mf do not differ, although the ranges do in some cases (Table H in S1 File). These results suggest that while there is some loss in model performance by fitting using the new hindcasting procedure, the procedure is still reliable to use in cases where no baseline data is available.

### Hindcasting baseline conditions in sites without pre-control data

With the confirmation that the hindcasting procedure can reasonably estimate baseline conditions, we calibrated the model to the three sites where no baseline data was available (Seri, Gbuwhen, and

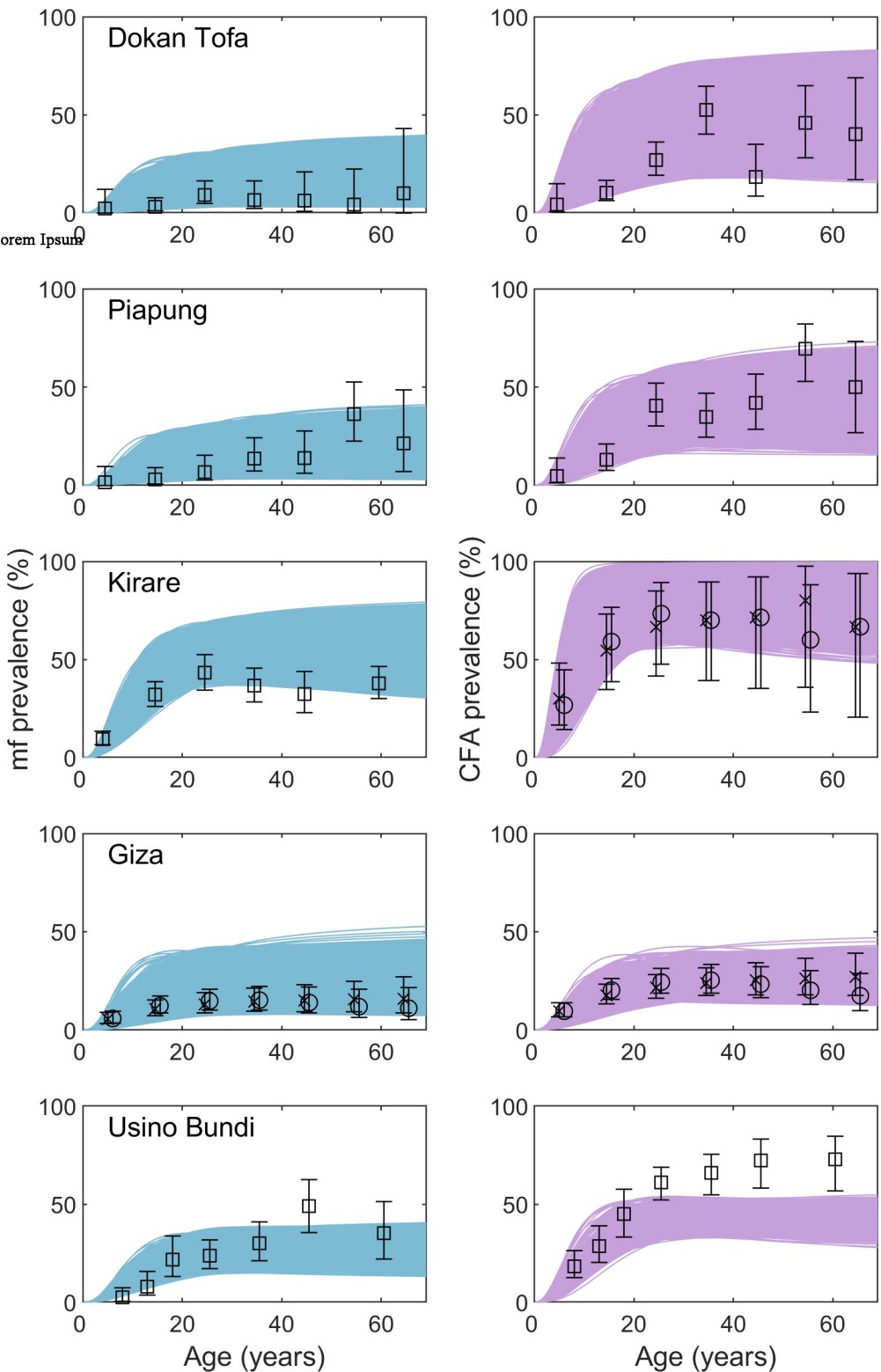

**Fig 2. Model hindcasts of baseline mf and CFA infection prevalence compared to observed survey data.** Each row of plots give the mf (left, blue curves) and CFA (right, purple curves) age prevalence for the given site at baseline. The curves represent the model-predicted infection by age compared to the observed (squares) or derived (plateau as crosses and convex as circles offset to visualize differences) age infection profiles. For all sites except CFA in Usino Bundi, the ensemble of curves passed through all of the data points indicating an accurate estimate of the baseline prevalence. Monte-Carlo p-values for assessing goodness of fit were >0.9 in each case.

**Table 2. Entropy of model fits to baseline data compared to model hindcasts of baseline.**

| Village | Entropy of model-predicted baseline mf prevalence | | Relative Information loss (%) |
|---|---|---|---|
| | Direct fit to baseline data | Hindcasted baseline | |
| Dokan Tofa | 9.6 | 13.0 | -35 |
| Piapung | 9.9 | 12.6 | -26 |
| Kirare | 9.0 | 13.3 | -43 |
| Giza | 9.0 | 10.0 | -11 |
| Usino Bundi | 9.3 | 9.6 | -2 |

Maiganga). The model fits to the 2003 mid-MDA age profile data are given Fig 3 and demonstrate that the model was able to accurately capture the infection profiles after many years of treatment had already occurred (Fig 1, calibration step 3). The selected parameter vectors from this fitting procedure were used to reconstruct the baseline mf and CFA age prevalence patterns (Table 3, Fig D in S1 File). The hindcasts suggest that the baseline mf prevalence was 48% (95% CI: 44–52%) in Seri, 30% (95% CI: 16–47%) in Gbuwhen, and 45% (95% CI: 33–52%) in Maiganga. Similarly, the model-estimated CFA prevalence at baseline was 73% (95% CI: 61–89%) in Seri, 77% (95% CI: 60–91%) in Gbuwhen, and 50% (95% CI: 18–69%) in Maiganga. Note that the relationship between mf and CFA predicted in Maiganga is not as we would expect. CFA prevalence was sometimes estimated to be lower than mf prevalence which is atypical and is reminiscent of the results seen for Usino Bundi where the model-estimated CFA prevalence was also lower than observed.

## Site-specific breakpoints calculations

We calculated LF mf breakpoints at both the ABR and the model-estimated TBR given that the value at TBR in sites where VC was used is required when modelling timelines to elimination in these settings [10]. The mf breakpoint representing a 95% elimination probability was calculated using an empirical inverse cumulative density function, and the site-specific values are given in Table 4. The values at TBR are statistically significantly higher than those at ABR (Kruskal-Wallis tests for each site, p-values < 2.2 e-16). The estimated mf breakpoint distributions also differed significantly between sites, indicating the existence of important between-site transmission heterogeneities (Kruskal-Wallis tests for breakpoints at ABR and TBR, p-values < 2.2 e-16). Practically, these differences were found to be more meaningful at TBR than at ABR and, in both cases, indicate that LF transmission thresholds in reality could be markedly lower than the WHO-recommended 1% mf threshold.

## Modelling the impact of site-specific MDA and vector control interventions

We used the site-specific transmission models to simulate the impact of the observed MDA and VC interventions in each site. Fig 4 shows the model-predicted mf and CFA prevalence over time compared to the longitudinal survey data collected in each site. The model predictions accurately reflect the observed changes in prevalence over time, further validating that the new model calibration approach developed in this study can accurately select site-specific parameter vectors based solely on data from a mid-MDA survey.

## Elimination and resurgence probabilities

The probabilities of elimination and resurgence are important metrics for evaluating the success of an elimination program. We calculated these values for each site given two scenarios: 1)

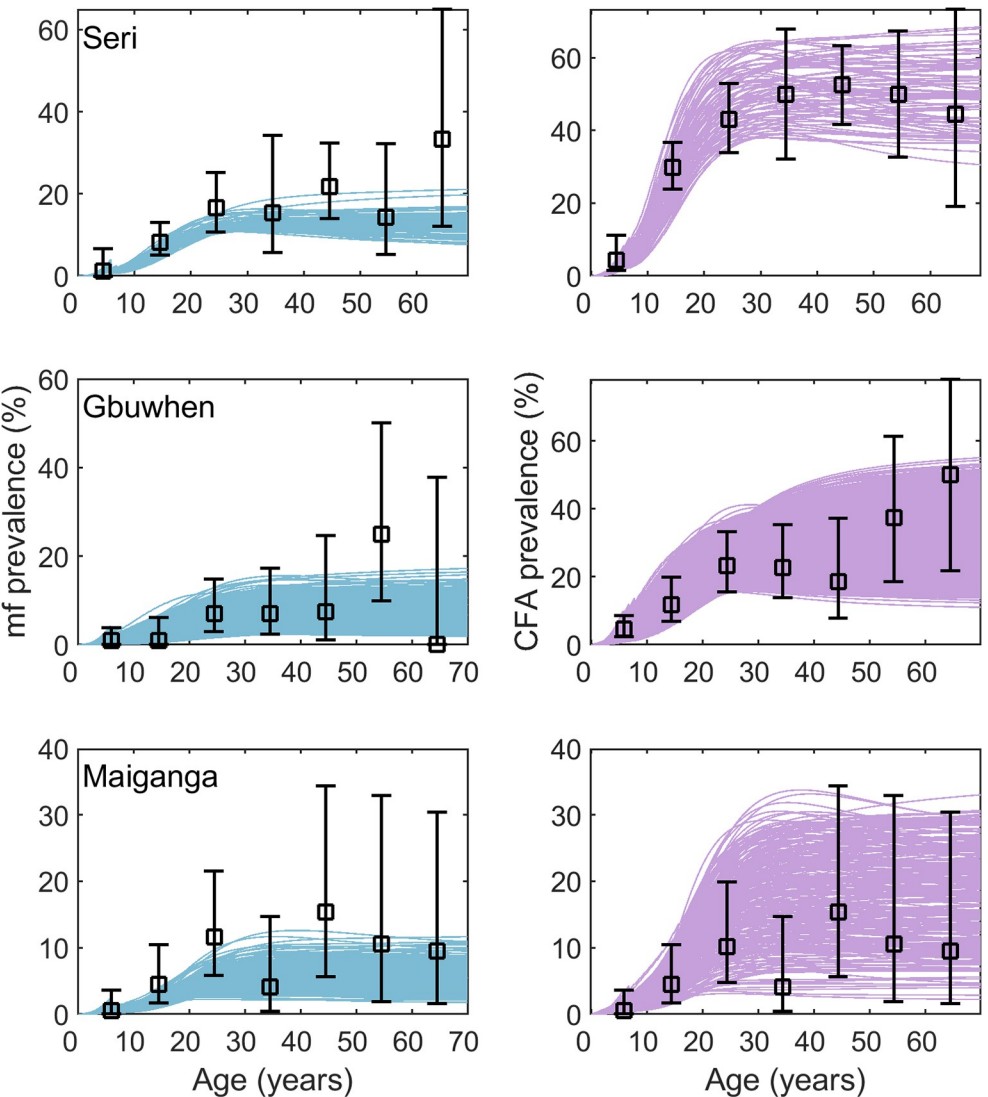

**Fig 3. Model fits to 2003 mid-MDA mf and CFA survey data.** Each row of plots give the mf (left, blue curves) and CFA (right, purple curves) age prevalence fits for the given site to data from the first LF survey in 2003 (after years of MDA for onchocerciasis had already occurred). The curves represent the model-predicted infection by age compared to the observed age infection profiles (black points with 95% confidence intervals).

**Table 3. Model-predicted age-stratified mf and CFA prevalence (median and 95% CI) for Seri, Gbuwhen, and Maiganga, Nigeria in 2003[1].**

| Age group | Seri | | Gbuwhen | | Maiganga | |
|---|---|---|---|---|---|---|
| | mf prevalence (%) | CFA prevalence (%) | Mf prevalence (%) | CFA prevalence (%) | Mf prevalence (%) | CFA prevalence (%) |
| 0–9 | 11.2 (7.3–16.3) | 15.9 (9.1–27.3) | 5.8 (2.3–13.5) | 15.7 (8.4–27.7) | 10.5 (5.5–18.5) | 8.9 (2.7–18.8) |
| 10–19 | 49.4 (41.1–56.7) | 61 (44.8–80.7) | 29.5 (14.1–50.8) | 61.7 (42.2–80.9) | 46.7 (31.6–56.1) | 37.8 (13–58.2) |
| 20–29 | 62 (57.4–66.9) | 73.2 (59.8–88.3) | 40.7 (23.3–61.1) | 76.1 (58.5–90.7) | 57.3 (43.6–65.8) | 49 (16.9–67.4) |
| 30–39 | 63.2 (57.8–70) | 74 (62.1–88.5) | 43 (26–63.3) | 78.6 (60.7–92.5) | 58.2 (42.3–68.8) | 49 (17.9–68.7) |
| 40–49 | 62.8 (53.7–71.7) | 73.8 (61.7–89.2) | 43.5 (26.4–64.1) | 79 (60.2–93.3) | 58.2 (42.3–69.6) | 49.2 (17.9–69.7) |
| 50–59 | 62.6 (50.6–72.6) | 73.7 (60.1–89.9) | 43.6 (26–65) | 79.2 (59.5–93.8) | 58.2 (41.6–70.6) | 48.7 (18–70.8) |
| 60+ | 62.3 (48.3–73.3) | 74 (57.3–90.4) | 43.7 (25.4–65.8) | 79.2 (58.6–94.2) | 58.3 (40.3–71.4) | 48.8 (18.1–71.2) |

[1] 2003 represents the first LF survey conducted in these sites, but many rounds of MDA had already occurred under the onchocerciasis control programs here. See Table 1 for complete context.

**Table 4. Site-specific mf breakpoints representing 95% elimination probability at either ABR or TBR.**

| Village | at ABR | at TBR |
|---|---|---|
| Dokan Tofa | 0.0097 | 0.3252 |
| Piapung | 0.0096 | 0.3477 |
| Kirare | 0.0086 | 0.7548 |
| Giza | 0.0010 | 0.2254 |
| Usino Bundi | 0.0102 | 0.4496 |
| Seri | 0.0072 | 0.7556 |
| Gbuwhen | 0.0080 | 0.5696 |
| Maiganga | 0.0054 | 0.4035 |

MDA is stopped once interventions have reduced the mf prevalence below the WHO 1% mf threshold, and 2) MDA is stopped once interventions have reduced the mf prevalence below the model-estimated site-specific mf breakpoint. In both cases, the use of bednets is assumed to continue beyond the MDA program in those sites where bednets were already present (i.e. all sites except Giza).

The results are given in Table 5. In six sites, the observed MDA program completed enough rounds of MDA (or more) to reach the WHO threshold. This resulted in high probabilities of elimination and low probabilities of resurgence. It is important, however, to note in this case that the use of VC protects against resurgence. The absence of this protective effect is seen in Giza where, despite having achieved the WHO threshold, there is only a 24% probability of elimination and consequently a 76% chance of resurgence. Conversely, all other sites that incorporated VC into their elimination programs were shown to have less than 10% probability of resurgence after MDA is stopped. However, if Giza were to continue MDA until reaching the model-estimated mf breakpoint, the risk of resurgence decreases to just 1%. These results highlight two important features of elimination dynamics: 1) that the WHO mf threshold may not be low enough to prevent resurgence after MDA is stopped, and 2) that supplementing MDA with VC can protect against resurgence. The results are also highly promising for the Nigerian sites. All five SVs studied here were predicted to have greater than 95% probability of elimination due to long term MDA and the use of LLINs. By contrast, Kirare, Giza, and Usino Bundi may have not completed enough rounds of MDA to ensure high probabilities of elimination.

## Discussion

Our fundamental focus in this study was to address the problem of how best to induce reliable parasite transmission models from data typically measured by control programs so that better policy-relevant forecasts of the impacts of interventions can be made. Data available from health agencies for model induction or discovery is often limited and may exhibit major gaps, as in the case of our Nigerian LF case study where many SVs do not have baseline parasite infection data available to inform initial conditions. Drawing on advances made in the field of computational knowledge discovery [8,19,51], particularly the notion that simulation models can serve as epistemic tools for reconstructing the natural systems under investigation [52], we posited that a computational system that would allow for both the generation of missing data as well as the identification of models based on prior knowledge of transmission structures and processes could facilitate solving this discovery problem. Here, we show specifically how a reconstructive computational system that combines data-assimilation with "common-sense" constraints to initialize models [8,27] can provide a powerful framework for generating the

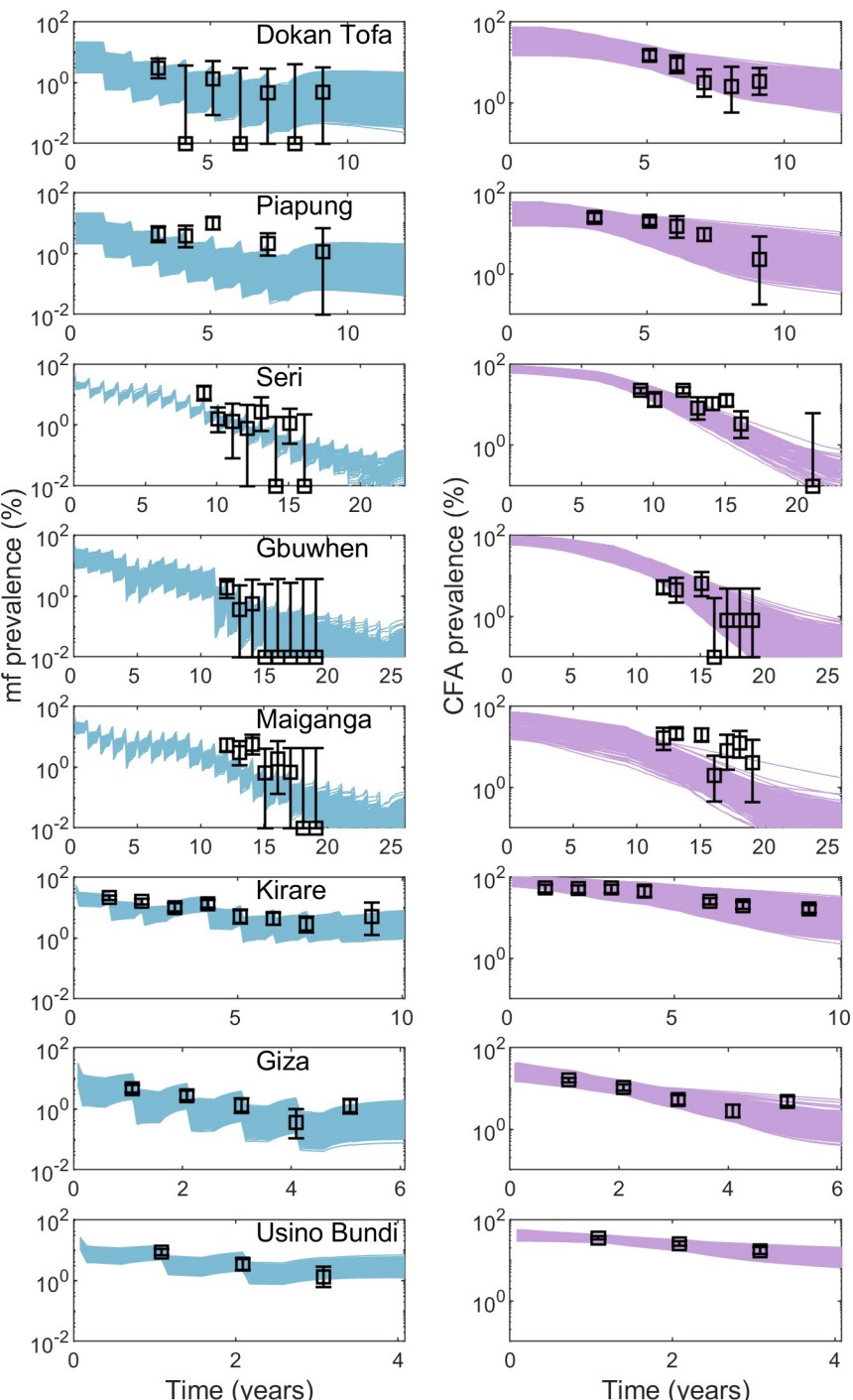

**Fig 4. Model forecasts of post-intervention mf and CFA prevalence compared to observed survey data.** Each row of plots give the model-predicted mf (left, blue curves) and CFA (right, purple curves) prevalence over time during the site-specific MDA program. Black points represent the surveyed prevalence with the corresponding 95% confidence intervals.

missing pre-control baseline information [21,28], which is critical for inducing locally-relevant infection models. Using the computational discovery system developed in this study, we successfully hindcasted the unknown pre-control LF prevalences of three SVs, estimated the

**Table 5. Probabilities of elimination and resurgence after the 5-year TAS period when MDA is stopped at 1) the WHO 1% mf threshold and 2) the model-predicted mf thresholds.**

| Village | Observed number of rounds of MDA | Presence of bednets[2] | WHO 1% mf threshold | | | Model-predicted site-specific mf threshold[1] | | |
|---|---|---|---|---|---|---|---|---|
| | | | MDA rounds required to reach threshold | Elimination probability[3] | Resurgence probability[4] | MDA rounds required to reach threshold | Elimination probability[3] | Resurgence probability[4] |
| Dokan Tofa | 7 | present | 7 | 100 | 0 | 11 | 100 | 0 |
| Piapung | 7 | present | 7 | 100 | 0 | 10 | 100 | 0 |
| Kirare | 8 | present | 14 | 91 | 9 | 15 | 91 | 9 |
| Giza | 5 | absent | 5 | 24 | 76 | 22 | 99 | 1 |
| Usino Bundi | 3 | present | 7 | 100 | 0 | 9 | 99 | 1 |
| Seri | 22 | present | 14 | 100 | 0 | 15 | 100 | 0 |
| Gbuwhen | 25 | present | 14 | 100 | 0 | 16 | 100 | 0 |
| Maiganga | 25 | present | 15 | 95 | 5 | 18 | 95 | 5 |

[1] The model-predicted site-specific thresholds represent the 95th percentile of the empirical inverse cumulative density function of the breakpoint estimates. The thresholds at TBR are applied for all sites except Giza where no vector control was used.

[2] Model forecasts assume that the existing vector control measures will continue after MDA is stopped.

[3] Elimination probability is calculated as the percentage of model forecasts that predict mf prevalence will continue to decrease after the 5-year TAS period.

[4] Resurgence probability is calculated as the percentage of model forecasts that predict mf prevalence will increase after the 5-year TAS period.

relevant locality-specific LF models, and generated new knowledge regarding breakpoints and durations of various interventions required to eliminate this major parasitic disease in different endemic settings. These results indicate that it will be beneficial for modellers to consider the construction and use of similar knowledge discovery platforms that take advantage of both 1) the information about a setting's particularities that is contained within partially available data and 2) prior information about the transmission dynamics of a particular disease in order to specify appropriate models for making the predictions needed to guide policy making [5].

Our first contribution in this study is to demonstrate how the new reconstructive computational data-model discovery system was able to identify local transmission models and reconstruct the pre-control endemic status of LF in a site by using generalized knowledge of filarial transmission dynamics, plausible baseline conditions, mid-MDA survey data, and details of historical interventions. The model hindcasts of the baseline mf and CFA prevalence matched the observed data well in the validation sites, highlighting the ability of this framework to reliably fill in data gaps across various LF endemic regions and under different intervention programs (Fig 2). Because the model was fitted primarily to monitoring data collected during MDA, this outcome is supportive of our previous conclusion that LF transmission parameters may be relatively stable over the course of typical MDA programs [21]. Such stability implies that the boundaries of local transmission dynamics are set by the initial conditions, which may largely remain unchanged, and that monitoring data can contain sufficient information for identifying key transmission characteristics. We also show that the estimated baseline mf and CFA prevalences in the three SVs where pre-control prevalence was unknown, namely Seri, Gbuwhen, and Maiganga (ie. before IVM treatments were begun for onchocerciasis) were relatively high (point estimates indicate 30–48% mf and 50–77% CFA prevalence, see Results). This assessment is consistent with the long history of IVM treatments experienced by these sites. It is also in agreement with results from independent empirical studies indicating that these SVs are from high prevalence local government areas (LGAs) in Nigeria and will require more rounds of treatment to interrupt transmission compared to other LGAs [34,37].

Together with the findings from the validated sites, these estimates highlight the utility of using models constrained with monitoring data for generating missing data pertaining to local baseline LF prevalence conditions in a given site. This constitutes a highly useful capability for disease management applications, since the discovery of information about pre-control prevalence in settings without such data is essential to policy makers for planning necessary interventions, determining the optimal frequency of monitoring activities, evaluating program progress, and applying for formal validation of elimination from the WHO [4,5,26,53].

This work has also provided additional insights regarding the value of data for refining parasite transmission models [13,16–18,22]. We found that there was a loss of information (as measured by the Shannon information index) when we calibrated the model to mid-MDA data and hindcasted the initial conditions as opposed to fitting the model directly to baseline data. The relative loss of information ranged from 2–43% (Table 2), but the model-predicted transmission breakpoints and the median expected timelines to achieve elimination were not notably different between the two calibration approaches (Fig C in S1 File and Table H in S1 File). This suggests that the observed loss of information, while statistically relevant, may not be practically meaningful. This conclusion complements previous findings that predictive performance of LF models is improved by calibrating the model to baseline and monitoring data together [18]. In Michael et al. (2018), for example, we show that data-informed models provide more precise predictions than unconstrained models and that it is valuable to also include post-intervention data in addition to baseline data to the model calibration procedure [18]. Taken together with the results of this study, we can conclude that, while baseline data may provide the most information for model calibration, monitoring data combined with the domain knowledge embedded in process-based models can both improve forecast horizons [14,23] and serve as reasonable substitute for identifying LF transmission dynamics in the absence of pre-control observations. This finding also may imply that computational irreducibility may not majorly affect the predictability of the LF models discovered using the monitoring data in this study [14].

However, it is important to note that model bias can still occur when initial conditions are not fully known. As highlighted by Beven (2007), when there is no empirical data to constrain the boundary conditions of a system, the results are dependent critically on modeling assumptions [54]. Here, we assumed limits on baseline prevalence and biting rates by implementing reasonable problem-space constraints, which can clearly influence the present model outcomes. Bayesian approaches, such as the BM methodology employed in this work, are highly advantageous in this scenario as the weight of the initially set subjective priors is reduced as the model is updated with more informative data [54]. Also addressing this concern about prediction bias is the fact that the identified models are able to make forecasts consistent with longitudinal monitoring data, which further increases confidence in model performance [55] (Fig 4). These findings indicate that the prediction bias of the LF models discovered here using only partially observed monitoring data is likely to be low; overall, coupled with the data generative ability of these models, the present results support the proposal by Dzeroski et al. (2007) for computational science to strive to make the most of the observations available rather than wait for ideal datasets that may never come [19]. This outcome should also serve as encouragement for programs to continue collecting data even if reports are incomplete as meaningful insights using models can still be made from the data that is available.

The intervention simulations carried out for each of the investigated study sites demonstrate how the site-specific models discovered from incomplete monitoring data using our computational approach are able to reliably estimate the baseline force of infection in order to simulate the impact of the MDA programs conducted in each study site. For all the present study sites, model forecasts of the interventions carried out in each site not only matched

temporal infection observations (Fig 4), but also provided information regarding when transmission interruption is likely to have occurred with high probability. This can be seen most strongly from the results pertaining to the three Nigerian SVs that do not have baseline data available (Seri, Gbuwhen, and Maiganga); our analysis indicates that there is a high probability (>95%) that the long-term MDA programs conducted in each of these SVs did indeed lead to elimination with a low risk of infection resurgence after stopping MDA (≤ 5%) (Table 5). These three SVs each conducted a total of 22–25 rounds of MDA (including the years of IVM monotherapy for onchocerciasis), but interventions are predicted to have successfully reduced mf prevalence below the WHO threshold after only 14–15 rounds of MDA and below the model-estimated site-specific mf thresholds after 15–18 rounds (Table 4, Table 5). These predicted timelines of transmission interruption are in agreement with the survey data from these sites that indicate that the mf prevalence was reduced to 0% by the end of the LF program in 2012, i.e. during the years following model-predicted transmission interruption (Table 1, Table 5, Fig 4). These results also indicate that more rounds of interventions were carried out than needed to achieve parasite elimination in these sites (at least 7–9 extra MDAs) (Table 5). In all three sites, 2–5 monitoring surveys had detected 0% mf prevalence before the IVM+ALB MDA for LF was stopped (Table 1), further supporting this conclusion.

Our assessment of elimination and resurgence probabilities also suggested a protective effect of VC against the reemergence of transmission after stopping MDA. In seven of the eight study sites (all but Giza), bednets were included in the control efforts and were assumed to continue after the MDA programs have ended. In these seven sites, high probabilities of elimination (> 90%) (and, consequently, low probabilities of resurgence) were predicted even after stopping MDA at 1% mf prevalence, which is notable because this threshold was previously found in modeling and field studies to be too high to guarantee elimination if all interventions are stopped (Table 5) [9,10,20,21,56,57]. On the contrary, in Giza, where bednets were not considered, 1% mf was not predicted to be sufficiently low to interrupt transmission as indicated by a high probability of infection resurgence after stopping annual treatments (76%). These results together suggest that the continued use of VC in the other seven sites had a protective effect against resurgence and that the continued use or introduction of VC after stopping MDA could lower the risk of resurgence considerably and safeguard the public health gains made by elimination programs. Previous modeling and field studies have also emphasized the value of VC for protecting against resurgence after stopping MDA [9,10,36,58]. In a study particularly relevant to the Nigerian scenario, Eigege et al. (2013) noted that eight years of MDA was not able to interrupt transmission in the vectors in Nasarawa and Plateau states, but that the incorporation of LLINs distributed by the malaria program was key for clearing the remaining *Wuchereria bancrofti* infection in mosquitoes [36].

We note that further refinement of our CFA model is necessary given that the model underestimated the prevalence of antigenaemia at baseline in Usino Bundi and during interventions in Maiganga (Fig 2, Fig 4, Table 3). In our current model, the sensitivity and specificity of the diagnostic tests are not considered and the apparent prevalence is assumed to represent the true infection rates [28]. Including diagnostic uncertainty for ICT is an important next step for estimating the true prevalence of CFA which may better guide the parameter choices describing the dynamics of antigen production in response to worms. It would also be beneficial to more fully investigate the functional relationship between worm and antigen dynamics, as we currently use a simple production-decay function to describe this process [9]. Datasets containing baseline and monitoring survey data for both mf and CFA indicators concurrently are sparse, which points to a need for increased data sharing opportunities through partnerships between modelers and program managers if models are to provide useful predictions. Another future direction for this work is to incorporate mosquito infection data into our modeling

study. Nigeria and TCC have collected extensive data on mosquito infection throughout their MDA programs which could provide new insights about the sensitivity of different indicators for detecting the interruption of transmission [34]. Due to the faster dynamics in the vector population owing to short mosquito lifespans, infection in vectors may give a more accurate reflection of ongoing transmission compared to mf and CFA indicators [9].

## Supporting information

**S1 File.**
(DOCX)

## Acknowledgments

We are grateful to the Nigerian LF Elimination Programme and staff of The Carter Center's Nigeria offices, whose hard work provided survey data for carrying out the calculations reported in this manuscript. MES gratefully acknowledges the fellowship support of this work by the Eck Institute for Global Heath, University of Notre Dame. A portion of the modelling work was carried out using the MATLAB Parallel Computing Toolbox made available on compute clusters of the University of Notre Dame's Center for Research Computing.

## Author Contributions

**Conceptualization:** Morgan E. Smith, Emily Griswold, Brajendra K. Singh, Edwin Michael.

**Data curation:** Morgan E. Smith, Emily Griswold, Emmanuel Miri, Abel Eigege.

**Formal analysis:** Morgan E. Smith, Brajendra K. Singh, Edwin Michael.

**Methodology:** Morgan E. Smith, Emily Griswold, Brajendra K. Singh, Edwin Michael.

**Project administration:** Edwin Michael.

**Supervision:** Edwin Michael.

**Writing – original draft:** Morgan E. Smith, Brajendra K. Singh, Edwin Michael.

**Writing – review & editing:** Morgan E. Smith, Emily Griswold, Brajendra K. Singh, Emmanuel Miri, Abel Eigege, Solomon Adelamo, John Umaru, Kenrick Nwodu, Yohanna Sambo, Jonathan Kadimbo, Jacob Danyobi, Frank O. Richards, Edwin Michael.

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
