## [Decision Letter · Decision Letter 0]

28 Jan 2020

Dear Prof. Michael,

Thank you very much for submitting your manuscript "Predicting lymphatic filariasis elimination in data-limited settings: a reconstructive computational framework for combining data generation and model discovery" for consideration at PLOS Computational Biology.

As with all papers reviewed by the journal, your manuscript was reviewed by members of the editorial board and by several independent reviewers. In light of the reviews (below this email), we would like to invite the resubmission of a significantly-revised version that takes into account the reviewers' comments.

We cannot make any decision about publication until we have seen the revised manuscript and your response to the reviewers' comments. Your revised manuscript is also likely to be sent to reviewers for further evaluation.

Sincerely,

Natalia L. Komarova

Deputy Editor

PLOS Computational Biology

Natalia Komarova

Deputy Editor

PLOS Computational Biology

Reviewer's Responses to Questions

**Comments to the Authors:**

Reviewer #1: The authors have constructed a detailed mathematical analysis of LF transmission in selected sites, engaging in a type of computational exploration which is billed as 'model discovery'. This is a well-established group in the field of LF modeling and have related their work to specific policy questions.

The heart of the presented novelty seems to be Figure 1, where we are told "site specific models" are identified. What does this mean? Does it mean the choice of appropriate parameters based on Table 4? Or does it mean statistical model selection? So little detail is provided that the claims made cannot be evaluated, let alone reproduced.

Selected rather minor comments:

Line 24. "Increasing recognition of the importance of mathematical models" is a dubious claim, as is the remainder of the sentence regarding the plausible-sounding idea that "useful" models "must accurately reflect" processes "pertaining to local infection dynamics". What does _accurately_ really mean in this context? The authors go on to say the models "must be identified", and that is unclear also. Does this refer to statistical identifiability?

Line 39. Claim that the model has provided "critical" information; this is tendentious; recommend omit.

Line 56. "knowledge discovery system'"; recommend omit. What knowledge? Are they talking about baseline values of prevalence?

Reviewer #2: General comments:

The paper presents interesting and useful methods and results that are very relevant to the current landscape of NTD elimination. However, the manuscript is written in a very dense and difficult to digest style. In a number of instances the reader is expected to follow up on multiple references at a time to disentangle single pieces of information vital to the reproducibility of the work. Having followed up on a number of these references it is not always clear what the authors are referring to, making the methods difficult to follow and substantiate. The sentence structure is also often long-winded and over-punctuated, with many sentences stretching over 5 or more lines of text.

The entire manuscript would benefit from rewording to include shorter and more digestible sentences. In addition, the authors should ensure clearer description of the modelling and statistical processes deployed, rather than relying on excessive referencing. Some of these can be including in supplementary but there are a number that would benefit from being included in the main text. This will ensure ease of understanding and reproducibility of the work described.

Specific comments:

Abstract: Excessively long sentences (some up to 5 lines long) make it difficult to read. Consider revising.

49-51 “For many neglected…” Sentence structure is muddled - please revise.

89-92: Consider revising sentence, ideally into two shorter sentences

96-98: Repetition of “data” and general sentence structure/length.

206-208: This doesn’t currently make sense when read through - what does “or any other” refer to here? Please reword.

210: Why are there so many references here? To allow ease of reproducibility you should provide one or two of the most relevant and specific references for the methods being employed. Using a clump of 6 references when describing a single method is inappropriate.

Line 245: How were parameter vectors selected to ensure community mf prevalence distribution was uniform? How did you select/reject samples?

Line 252: Please expand on what this “pre-defined criteria of fit” is

Line 256: Does “single mid-MDA survey” mean a single time point? Why was only a single time point used?

Testing sites (Dokan Tofa, Piapung, Kirare, Giza and Usino Bundi) have no age-stratified mid-MDA data, the age profile has to be derived, so the model is essentially different in these sites than in Seri, Gbuwhen and Maiganga. Make this clearer, how does this impact validation?

273-281: The pruning method described here doesn’t appear to align with any described in the provided reference [47]. I may have missed it (feel free to point it out if so) but if not then please provide a suitable reference to confirm legitimacy of these pruning method.

297-299: How was the reduction in biting rate calculated? What is the functional relationship between ITN and LLIN coverage/efficacy and biting rate? You would expect ITN and LLIN usage to reduce the biting rate, but also reduce the prevalence of infection in the mosquito population, which does not appear to be considered here. The effect is commonly considered to be quadratic (repelling/preventing feeding slows disease acquisition and transmission).

312-313: Waning effect of bednets?

Recrudescence is not a suitable word as used in this manuscript. In a medical sense, recrudescence is the return of detectable symptoms in a patient whose blood stream infection has previously been at such a low level as not to be clinically demonstrable or cause symptoms, it does not refer to the re-establishment of disease at a population level. Resurgence would be a more suitable word.

314-319: How are marginally positive and negative slopes dealt with that could simply represent a maintenance of low-levels of infection rather than a trend towards elimination or resurgence?

323: Reference for Bayesian melding framework should be repeated here.

324-328: Is this supposed to describe the Bayesian melding framework? It sounds like basic approximate bayesian computation. Also the “binary criteria of fit” is not described anywhere in the manuscript. It is mentioned, with reference, in line 252 but not elaborated on what this actually means. This is a vital part of the methodology for reproducibility, please specify explicitly within the manuscript.

332: “fit” should be “fitted”

Figure 1: It’s very hard to see the crosses and circles, with their respective error bars, where they are overlaid. Also the area covered by the model-derived curves is most of the space in a number of plots (e.g. Dokan Tofa CFA). This doesn’t represent an accurate estimate of baseline prevalence, simply that the observed baseline prevalence lies within the very wide range of possibilities predicted by the model. An estimated baseline mf prevalence of between 0 and up to 50% in some age categories (Dokan Tofa, Giza, Piapung) isn’t particularly informative.

Table 3: Is this baseline prevalence? Specify in title.

Table 4: Is ABR here calculated in the complete absence of vector control or is this assuming the low coverage of vector control assumed in the model? Also a number of these (Dolan Tofa, Piapung, Kirare, Usino Bundi, Seri, and Gbuwhen) are within reasonable error of 0.01 mf (1% prevalence), or at least could not be called “markedly lower” as described in line 414. This is reflected by the very low (or zero) resurgence probabilities presented for these settings in Table 5. The highest resurgence probability is seen in Giza, where no bednets are present, which implies to me that the ABR presented in Table 4 may include local bednet usage. This is misleading as it suggests that transmission is naturally higher in Giza, but may mainly be due to a lack of bednet usage.

No rigorous statistical tests of goodness of fit are used in the methods, please justify.

Line 498: This is the range of point estimates with no uncertainty, could be misleading. Including uncertainty these are actually 16-52% mf and 18-91% CFA

**Have all data underlying the figures and results presented in the manuscript been provided?**

Reviewer #1: No: I can't tell where the data shown in Supplement Figure 1 are.

Reviewer #2: Yes

PLOS authors have the option to publish the peer review history of their article (what does this mean?). If published, this will include your full peer review and any attached files.

Reviewer #1: No

Reviewer #2: No
---

## [Decision Letter · Decision Letter 1]

12 May 2020

Dear Prof. Michael,

We are pleased to inform you that your manuscript 'Predicting lymphatic filariasis elimination in data-limited settings: a reconstructive computational framework for combining data generation and model discovery' has been provisionally accepted for publication in PLOS Computational Biology.

Best regards,

James Lloyd-Smith

Associate Editor

PLOS Computational Biology

Natalia Komarova

Deputy Editor

PLOS Computational Biology

Reviewer's Responses to Questions

**Comments to the Authors:**

Reviewer #2: The manuscript has been much improved.

**Have all data underlying the figures and results presented in the manuscript been provided?**

Reviewer #2: Yes

PLOS authors have the option to publish the peer review history of their article (what does this mean?). If published, this will include your full peer review and any attached files.

Reviewer #2: No

---

## [Editor Report · Acceptance letter]

13 Jul 2020

PCOMPBIOL-D-19-01845R1 

Predicting lymphatic filariasis elimination in data-limited settings: a reconstructive computational framework for combining data generation and model discovery

Dear Dr Michael,

I am pleased to inform you that your manuscript has been formally accepted for publication in PLOS Computational Biology. Your manuscript is now with our production department and you will be notified of the publication date in due course.

With kind regards,

Sarah Hammond
